# Mitigate Catastrophic Remembering via Continual Knowledge Purification for Noisy Lifelong Person Re-Identification

## ABSTRACT

Current lifelong person re-identification (LReID) methods focus on tackling a clean data stream with correct labels. When noisy data with wrong labels are given, their performance is severely degraded since the model inevitably and continually remembers erroneous knowledge induced by the noises. Moreover, the well-known catastrophic forgetting issue in LReID becomes even more challenging since the correct knowledge contained in the old model is disrupted by noisy labels. Such a practical noisy LReID task is important but challenging, and rare works attempted to handle it so far. In this paper, we initially investigate noisy LReID by proposing a Continual Knowledge Purification (CKP) method to address the catastrophic remembering of erroneous knowledge and catastrophic forgetting of correct knowledge simultaneously. Specifically, a Cluster-aware Data Purification module (CDP) is designed to obtain a cleaner subset of the given noisy data for learning. To achieve this, the label confidence is estimated based on the intra-identity clustering result where the high-confidence data are maintained. Besides, an Iterative Label Rectification (ILR) pipeline is proposed to rectify wrong labels by fusing the prediction and label information throughout the training epochs. Therefore, the noisy data are rectified progressively to facilitate new model learning. To handle the catastrophic remembering and forgetting issues, an Erroneous Knowledge Filtering (EKF) algorithm is proposed to estimate the knowledge correctness of the old model, and a weighted knowledge distillation loss is designed to transfer the correct old knowledge to the new model while excluding the erroneous one. Finally, a Noisy LReID benchmark is constructed for performance evaluation and extensive experimental results demonstrate that our proposed CKP method achieves state-of-the-art performance.

## CCS CONCEPTS

• **Computing methodologies** → **Object identification**; • **Information systems** → **Information retrieval**.

## KEYWORDS

Lifelong Person Re-Identification, Noisy Learning

## 1 INTRODUCTION

Person re-identification (ReID) [1, 22] is a classical multimedia task that has been thoroughly investigated in stationary scenes [1, 4, 10]. Recently, lifelong person re-identification (LReID) [6, 27, 28, 40], aiming to continually learn from the practical non-stationary data stream, has drawn increasing research attention. Nevertheless, existing LReID methods simply assume that the training data are all correctly annotated [27, 34]. In realistic scenarios, training labels are often inevitably noisy due to inaccurate person detection or annotation errors [5, 47, 48] which can hinder the stationary ReID performance severely [44, 51].

The LReID models are even more vulnerable to label noise, which refers to the mislabeled data in Figure 1 (a). As shown in Figure 1 (b), when noisy data are given for learning, the LReID model learns from erroneous identity-affinity supervision [45], leading to significant performance degradation. Besides, even though the model has

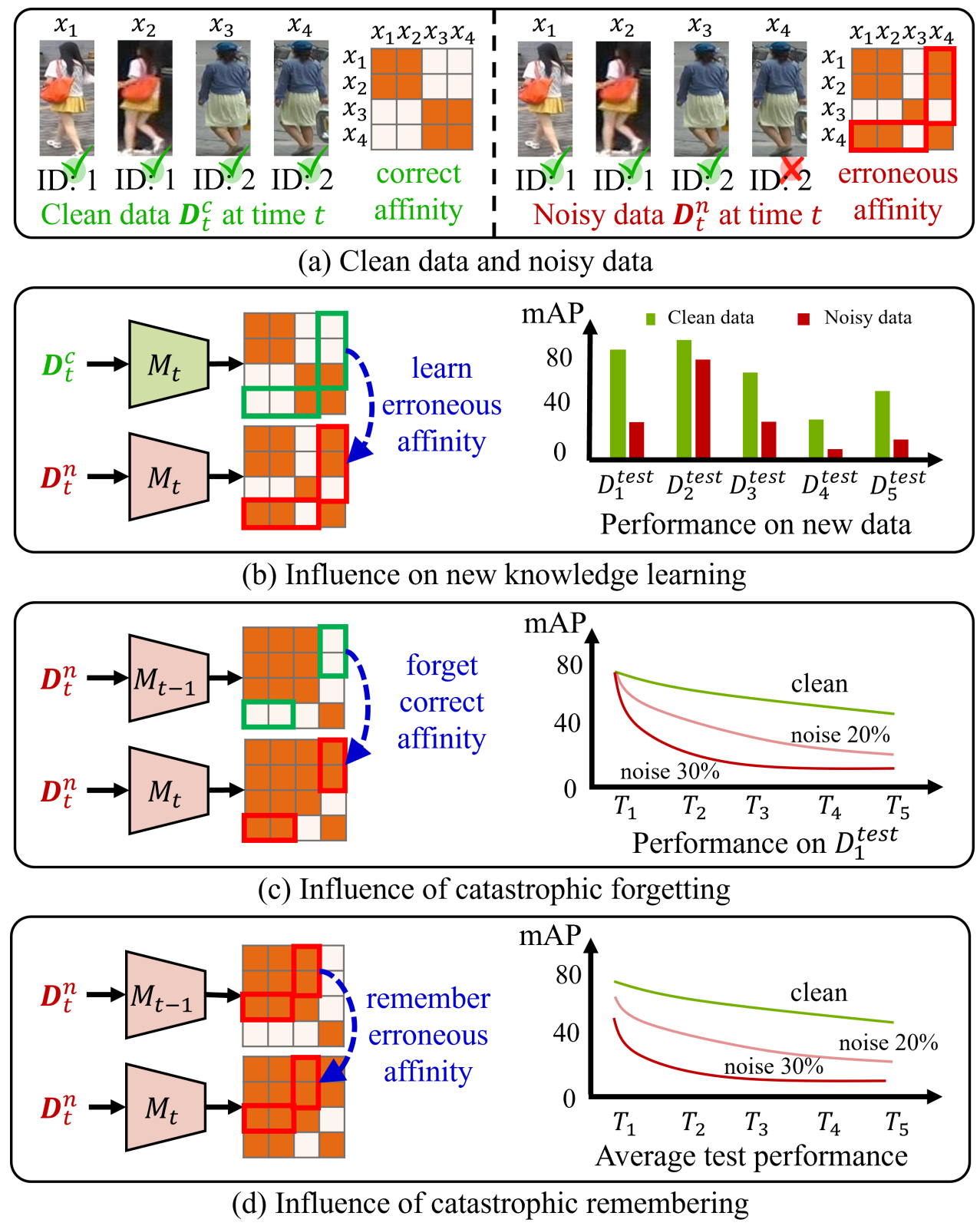

**Figure 1: (a) Noisy data contains wrong labels which introduce erroneous inter-instance affinity knowledge. During LReID, noisy data not only (b) influences the knowledge learning of new datasets but also (c) exacerbates the forgetting of correct historical knowledge. Besides, (d) the learned erroneous knowledge is remembered by the new model $M_t$ which severely hinders the LReID performance. The experiment results are obtained from the latest LReID method [42].**

acquired correct knowledge from historical data in Figure 1 (c), label noises in the new data can introduce erroneous information, overwriting the correct knowledge with erroneous knowledge [32, 54], thereby exacerbating catastrophic forgetting during new model learning. Additionally, as illustrated in Figure 1 (d), since the old model inevitably remembers fatal erroneous knowledge of historical noisy data, the anti-forgetting strategy aiming to maintain output consistency between the new and old models can cause erroneous knowledge accumulation and impede the acquisition of correct knowledge. These issues can significantly hinder the performance of the latest LReID method [42]. As presented in Figure 1 (b), the histogram displays the performance drop when learning a dataset with clean or noise labels respectively. The curves in Figure 1 (c) depict the performance trend on the first dataset, where the model learns with clean labels initially and subsequently adapts to datasets with varying noise ratios. Figure 1 (d) shows the average performance on all learned datasets with different noise ratios. As can be seen, label noise is a crucial and challenging problem for LReID.

Recently, several label noise learning (LNL) techniques have been explored to settle the noisy ReID and classification tasks [12, 47, 48]. They primarily rely on identity prediction results [47] or loss regularization strategies [48, 49] to mitigate the influence of noisy data [23, 36, 44]. However, these approaches neglect the disparity between the prediction/loss and the actual identity distribution [37]. As a result, noisy samples near the distribution boundary tend to exhibit prediction scores or regularization effects that are indistinguishable from those of clean samples. Thus, some wrongly labeled samples can easily be confused with clean ones, leading to the accumulation of erroneous knowledge during learning. Therefore, when directly applying these LNL methods to Noisy LReID without considering the characteristics of LReID, the issues illustrated in Figure 1 remain critical.

In this paper, we initially investigate this challenging noisy LReID task and propose a novel method named Continual Knowledge Rectification (CKR) to handle the catastrophic remembering and forgetting issues. Our approach can not only adaptively rectify the noisy samples to ensure the learning of correct knowledge but also actively forget the remembered erroneous knowledge from the old model. Specifically, a Cluster-aware Data Purification module (CDP) and an Iterative Label Rectification pipeline (ILR) are proposed in our CKR to achieve high-quality clean data. CDP can adaptively select the clean samples for new model training by estimating the label confidence from intra-identity clustering, thereby settling the problems in Figure 1 (b) and (c). Instead of simply discarding the wrongly labeled samples, the proposed ILR aims to fully utilize them by rectifying their labels along with model learning so that these data can be recollected by CDP for reuse. Furthermore, to actively forget the erroneous old knowledge, an Erroneous Knowledge Filtering algorithm (EKF) is proposed to estimate the knowledge correctness of the old model outputs, and a weighted knowledge distillation loss is designed to transfer the correct old knowledge to the new model while excluding the erroneous one. Thus, the erroneous knowledge remembering issue in Figure 1 (d) could be greatly mitigated. To evaluate the performance of our method, a Noisy LReID Benchmark (NLReID) is proposed inspired by the existing LReID and LNL benchmark configurations [27, 47]. Extensive experimental results under various noisy conditions demonstrate the superiority of our CKR model.

In summary, the contributions of this work are three-fold: (1) We provide a pioneer investigation on the important and challenging Noisy LReID task, and a comprehensive Noisy LReID benchmark (NLReID) is proposed for the evaluation of existing methods. (2) To handle the catastrophic remembering and forgetting issues, a novel Continual Knowledge Rectification (CKR) method is proposed. A Cluster-aware Data Purification module and an Iterative Label Rectification pipeline are designed to obtain cleaner training data for correct new knowledge learning and mitigating erroneous new knowledge acquisition. Besides, an Erroneous Knowledge Filtering algorithm is developed to actively forget erroneous old knowledge and ensure correct new knowledge remembering. (3) Extensive experiments demonstrate that our CKR achieves state-of-the-art Noisy LReID performance, and the proposed method can be readily integrated with the latest LReID or LNL approaches to further improve the performance in the noisy LReID scenario.

## 2 RELATED WORK

### 2.1 Lifelong Person Re-Identification

Lifelong person re-identification (LReID) [27, 40] aims to train a ReID model with non-stationary data, improving the model's adaptability to various conditions. Existing LReID works [6, 27–29, 34, 40, 50] primarily focus on alleviating the catastrophic forgetting problem, which indicates that the performance of the model on historical data is degraded greatly when the new data is learned [13, 35, 38, 46]. Nevertheless, these methods assume that the training data are all correctly annotated [27, 40]. However, in real scenarios, the training data labels are often noisy due to inaccurate person detection or annotation errors [3, 5, 47]. Such a Lifelong Person Re-Identification with the Noisy Label (Noisy LReID) scenario is more challenging since not only the correct knowledge catastrophic forgetting exacerbated due to erroneous new knowledge continually overwriting the correct old knowledge, but also catastrophic remembering [15, 52] issues occur as the erroneous knowledge from different domains accumulates, resulting in degraded performance on new domains.

### 2.2 Label Noise Learning

Label Noise Learning (LNL) has drawn much research attention in recent years [18, 33, 55]. Most existing LNL methods rely on identity prediction [12, 44, 47] or loss regularization strategies [23, 48, 49] to filter noisy data or accomplish noise-robust learning. For example, LCNL [45] adopts Gaussian Mixture Model (GMM) [36, 44] to model the loss distribution and select the unreliable samples. CORE [47] introduces a regularization loss to mitigate the influence of the label on high-confidence prediction. However, existing works reveal that there is a discrepancy between the prediction score/loss and the actual identity distribution [37]. Specifically, the noisy samples around the distribution boundary tend to exhibit indistinguishable prediction scores or regularization effects from the clean ones. Thus, these samples can easily be confused with the wrong labels, resulting in inaccurate filtering or invalid regularization. Therefore, the learned models in these methods contain considerable erroneous knowledge, and the catastrophic remembering problem in the Noisy LReID scenario is still critical.

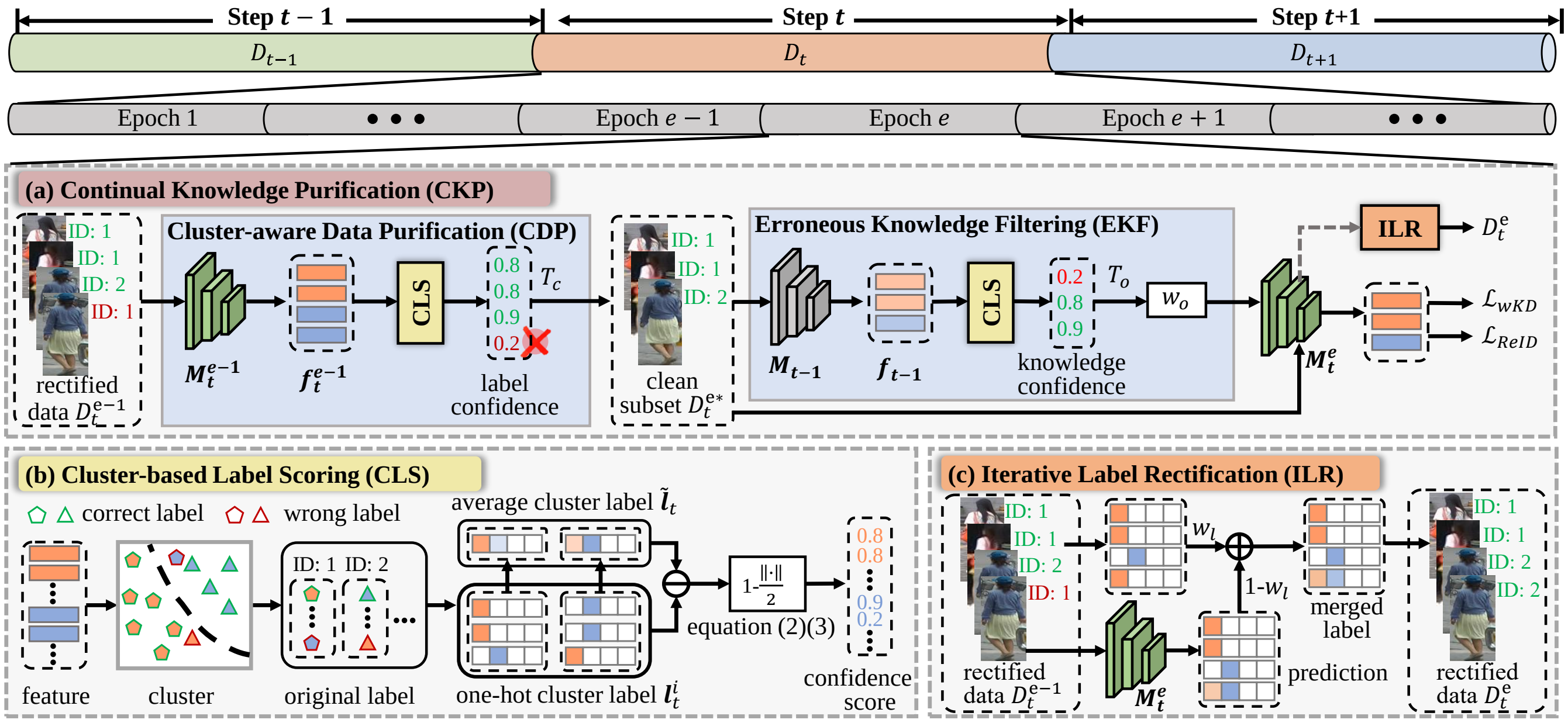

**Figure 2: Given the noisy training dataset $D_t$ at the $t$-th LReID step, our CKR updates the rectified noisy data $D_t^e$ and model $M_t^e$ along the training epoch $e$. The CDP aims to obtain a clean subset $D_t^{e*}$ for the learning of $M_t^e$. Besides, the EKF aims to filter the features containing erroneous old knowledge. Finally, the ILR proposes to rectify the noise labels by fusing the learning knowledge and label information.**

## 2.3 Lifelong Learning with Label Noise

Lifelong (Continual) Learning with Label Noise problem has yet to gain widespread attention and existing solutions focus on classification task [9, 14, 16]. These methods rely on filtering and retaining historical exemplars to address the catastrophic forgetting issue during lifelong learning. However, as human images are highly privacy-sensitive data, retaining historical exemplars is not feasible in many actual applications [27, 34]. Therefore, in this paper, we provide a pioneer investigation on the Noisy LReID problem, the catastrophic forgetting of correct knowledge and the catastrophic remembering of erroneous knowledge under such a scenario is thoroughly discussed, and a novel exemplar-free Noisy LReID method showing state-of-the-art performance is proposed.

# 3 CONTINUAL KNOWLEDGE PURIFICATION FOR NOISY LREID

## 3.1 Problem Definition and Formulation

Noisy lifelong person re-identification (Noisy LReID) aims to continually learn from a stream of $T$ ReID datasets $\mathcal{D}^{tr} = \{D_t\}_{t=1}^{T}$, each containing a certain ratio of noisy labels. The effectiveness of the final model is evaluated on the clean test sets $\mathcal{D}^{te} = \{D_t^{te}\}_{t=1}^{T}$ corresponding to each domain, to evaluate the new knowledge acquisition and anti-forgetting capacity of the model. Besides, a series of additional $U$ clean test sets $\mathcal{D}^{un} = \{D_t^{un}\}_{t=1}^{U}$ are tested to evaluate the generalization of the models on unseen domains. In this paper, the model learned after training step $t$ is denoted as $M_t$ and the intermediate model after each training epoch $e$ is denoted as $M_t^e$. The parameters initial model $M_t^0$ is copied from $M_{t-1}$.

## 3.2 Overview

As is shown in Figure 2, given the noisy training dataset $D_t$ at training step $t$, our overall approach generates a rectified dataset $D_t^e$ and model $M_t^e$ at the training epoch $e$, where $D_t^0$ is initialized with original noisy data $D_t$. The proposed framework consists of three key components, *i.e.*, Cluster-aware Data Purification (CDP), Erroneous Knowledge Filtering (EKF), and Iterative Label Rectification (ILR). Specifically, the CDP module aims to estimate label confidence for each instance and generate a clean subset $D_t^{e*}$ to ensure accurate new data learning. Then, EKF is employed to estimate the knowledge correctness of the old model features, so that the erroneous knowledge could be actively forgotten and the correct new knowledge could be consolidated. Besides, at the end of the $e$-th epoch, the IRL pipeline is adopted to rectify the noisy labels by fusing the model prediction and label information, resulting in the rectified dataset $D_t^{e+1}$ for subsequent epochs. Since the label confidence estimation function serves as a crucial component for the proposed CDP and EKF modules, we introduce the proposed Cluster-aware Label Scoring strategy first and depict the CDP, EKF, and ILR designs sequentially.

## 3.3 Cluster-aware Label Scoring

In this work, we propose to utilize clustering [43] technology to gather instances with shared characteristics to achieve reliable label confidence estimation. Specifically, given the noisy dataset $\{(x_i, y_i)\}_{i=1}^{N^t}$ with $N_t$ images $x_i$ and corresponding labels $y_i$, the extracted features are $\{f_i\}_{i=1}^{N_t}$. The DBSCAN algorithm [31] is adopted to generate clusters with different shared characteristics and each

instance is assigned a cluster label $\tilde{y}_i \in \{1, 2, ..., N_c\}$ where $N_c$ is a cluster-aware identity. Note that the outliers during the clustering process are collected as an extra cluster whose label is set to $N_c$. Then, we generate one-hot embedding $\boldsymbol{l}_i \in \mathbb{R}^{N_c}$ for all instances.

Then, to bridge the connection between the annotated label and the generated cluster label, an annotation-aware average cluster label $\bar{\boldsymbol{l}}_i$ for each instance $x_i$ is calculated by

$$\bar{\boldsymbol{l}}_i = \frac{1}{n_t^i} \sum_{j=1}^{N_t} \delta(y_i, y_j) \boldsymbol{l}_i, \tag{1}$$

where $\delta(y_i, y_j)$ is a sign function that outputs 1 and 0 when $y_i = y_j$ and $y_i \neq y_j$ respectively. $n_t^i = \sum_{j=1}^{N_t} \delta(y_i, y_j)$ is the instance number of the identity $f_i$ belonging to. Equation (1) indicates that given $x_i$ with annotated label $y_i$, obtain the average cluster label of all instances $x_j$ annotated with label $y_i$. Therefore, $\bar{\boldsymbol{l}}_i$ is shared across instances with the same annotated label and reflects the overall value of each annotated label in the cluster label space.

To quantify the label confidence of each instance $x_i$, the label distance $d_i$ is defined as the squared L2 norm between $l_i$ and $\bar{\boldsymbol{l}}_i$, represented as:

$$d_i = ||\boldsymbol{l}_i - \bar{\boldsymbol{l}}_i||_2^2, \tag{2}$$

where $d_i$ measures the disparity between the cluster label $\tilde{y}_i$ and the annotated identity centers. Note that $d_i \in [0, 2)$ and the $d_i$ values of outliers during clustering are adjusted to 2. This adjustment is made because the outliers exhibit minimal resemblance to other instances and thus possess the lowest confidence.

Then the annotated label confidence score $s_i$ is calculated by

$$s_i = (2 - d_i)/2, \tag{3}$$

where $s_i \in [0, 1]$ with higher values indicating greater trustworthiness of the annotated label.

**Discussion:** Existing label confidence scoring methods primarily utilize the Gaussian Mixture Model (GMM) to model the noise distribution for clean data selection [12, 45]. However, since there is a discrepancy between the model loss and the actual identity distribution [37], GMM can reserve many noisy samples, thereby limiting the new knowledge acquisition and correct knowledge anti-forgetting capacity (Figure 1 (a)(b)). However, the elaborately designed CLS strategy can effectively mine fine-grained inter-instance similarity to evaluate identity coherence across instances, therefore the intra-identity distribution is fully modeled and utilized to enhance the reliability of label confidence estimation results.

## 3.4 Cluster-aware Data Purification

As is shown in Figure 2, given the input dataset $D_t^{e-1} = \{(x_i, y_i)\}_{i=1}^{N_t}$ at $e$-th epoch of training step $t$, the previous epoch learned model $M_t^{e-1}$ is used to process the images and obtain the features $\{f_t^i\}_{i=1}^{N_t}$. Then the CLS strategy is adopted to process $\{f_t^i\}_{i=1}^{N_t}$ to get the label confidence $s_i$ of each image $x_i$.

Then, we remove the data of low-confidence labels with a confidence threshold $T_c$ and obtain the clean subset $D_t^{e*}$:

$$D_t^{e*} = \{(x_i, y_i)\}_{i=1}^{N_t^*}, \tag{4}$$

where each instance $x_i$ in $D_t^{e*}$ has the $s_i$ higher than $T_c$ and $N_t^*$ is the selected instance number.

## 3.5 Erroneous Knowledge Filtering

The knowledge distillation [7] strategy is a widely-used anti-forgetting approach adopted by the existing LReID methods [6, 27, 34, 40]. Despite its knowledge consolidation capacity, such a strategy can lead to erroneous knowledge accumulation and even mislead the learning of the new data, as shown in Figure 1 (d).

Therefore, in this section, we aim to filter the features of samples that reflect the learned correct knowledge of old model $M_{t-1}$ and discard the features that contain erroneous knowledge contained in $M_{t-1}$. Specifically, given the clean subset $D_t^{e*}$, we utilize $M_{t-1}$ to process the $D_t^{e*}$ and the generated features $\{f_{t-1}^i\}_{i=1}^{N_t^*}$ are fed into the CLS (Section 3.3), where the obtained scores of each instance is named feature confidence $s_i^o$. Then a knowledge distillation weight $w_i^o$ is assigned to each instance $x_i$ by

$$w_i^o = \begin{cases} 0 & s_i^o \leq T_o \\ 1 & s_i^o > T_o \end{cases}, \tag{5}$$

where $T_o$ is the hyperparameter serving as the threshold of $s_i^o$. Notably, $w_i^o = 0$ indicates that the old knowledge can not correctly process the instance $x_i$, thus $f_{t-1}^i$ primarily contains erroneous old knowledge and should be discarded during knowledge distillation.

Then we proposed a weighted knowledge distillation loss $\mathcal{L}_{wKD}$ to ensure correct old knowledge transfer and active erroneous old knowledge forgetting. Considering there are primarily two kinds of knowledge distillation loss, *i.e.* logits-based and inter-instance relation-based. We design the $\mathcal{L}_{wKD}$ variants accordingly.

Specifically, as for logits-based knowledge distillation loss [24, 27], $w_i^o$ serve as the weight of each instance directly:

$$\mathcal{L}_{wKD-lgs} = w_i^o \mathcal{L}_{KD}(M_{t-1}(x_i), M_t^e(x_i)), \tag{6}$$

where $\mathcal{L}_{KD}$ is a ordinary loss function, *e.g.*, KL-diverigence [42], MSE [26]. As for the inter-instance relation-based knowledge distillation, given a batch of instances $\mathcal{B}$, a maximum subset $\mathcal{B}_o$ where each instance with $w_i^o = 1$ is selected to calculate the inter-instance relation loss:

$$\mathcal{L}_{wKD-rel} = \mathcal{L}_{KD}\left(\theta_r\left(M_{t-1}\left(\mathcal{B}_o\right), M_t\left(\mathcal{B}_o\right)\right)\right), \tag{7}$$

where $\theta_r$ is a relation evaluation function [42] and $\mathcal{L}_{KD}$ is a relation knowledge distillation loss [34].

## 3.6 Iterative Label Rectification

Although the above CDP and EKF modules could ensure the model learning the correct knowledge, the wrongly labeled data which can contain abundant information are discarded. To settle this drawback, we propose to rectify the annotated data iteratively along the model learning epochs, ensuring the correct knowledge learning and enhancing informative data utilization simultaneously.

Specifically, the label rectification is accomplished by

$$y_i^* = \text{argmax}\{\boldsymbol{y}_i * w_l + \hat{\boldsymbol{y}}_i * (1 - w_l)\}, \tag{8}$$

where $\boldsymbol{y}_i \in \mathbb{R}^{N_p}$ is a one-hot embedding generated from annotated label $y_i$ in $D_t^e$ and $N_p$ is the annotated person identity number. The label rectification weight $w_l$ aims to fuse the information of annotation and prediction. $\hat{\boldsymbol{y}}_i \in \mathbb{R}^{N_p}$ is the identity prediction vector generated by $M_t^e$. $y_i^*$ is utilized to replace $y_i$ in $D_t^e$ to obtain new dataset $D_t^{e+1}$.

**Table 1: Results under the Random Noise. † indicates the state-of-the-art LNL method is combined with the latest anti-forgetting strategy of LSTKC.**

| Metric | Type | Method | Market-1501 | | | CUHK-SYSU | | | DukeMTMC | | | MSMT17 | | | CUHK03 | | | Seen-Avg | | | Unseen-Avg | | |
|---|---|---|---|---|---|---|---|---|---|---|---|---|---|---|---|---|---|---|---|---|---|---|---|
| | | | 10% | 20% | 30% | 10% | 20% | 30% | 10% | 20% | 30% | 10% | 20% | 30% | 10% | 20% | 30% | 10% | 20% | 30% | 10% | 20% | 30% |
| mAP | LReID | LwF [24] | 49.2 | 29.1 | 19.2 | 66.2 | 53.7 | 47.7 | 19.0 | 9.8 | 7.5 | 3.6 | 1.9 | 1.3 | 17.3 | 8.9 | 6.1 | 31.1 | 20.7 | 16.4 | 35.4 | 25.6 | 20.2 |
| | | PatchKD [34] | **51.0** | 30.6 | 23.9 | 66.8 | 58.0 | 52.6 | 19.3 | 11.9 | 8.4 | 3.6 | 2.1 | 1.5 | 18.1 | 10.3 | 6.4 | 31.8 | 22.6 | 18.6 | 35.0 | 27.3 | 23.8 |
| | | KRKC [50] | 29.4 | 20.1 | 14.8 | 71.7 | 65.3 | 59.2 | 23.3 | 15.3 | 10.3 | 5.1 | 3.1 | 2.2 | 35.3 | 20.9 | 12.7 | 33.0 | 24.9 | 19.8 | 42.5 | 35.4 | 29.0 |
| | | DKP [17] | 43.2 | 28.1 | 20.4 | 78.4 | 71.0 | 66.7 | 34.8 | 22.5 | 15.4 | 12.8 | 7.7 | 5.5 | 21.2 | 10.7 | 6.6 | 38.1 | 28.0 | 22.9 | 47.4 | 37.6 | 32.5 |
| | | LSTKC [42] | 41.4 | 35.1 | 25.8 | 78.7 | 74.2 | 67.4 | 39.7 | 19.4 | 9.3 | 14.8 | 5.2 | 2.6 | 27.8 | 15.3 | 6.9 | 40.5 | 29.8 | 22.4 | 47.2 | 37.0 | 30.0 |
| | LNL | CORE [47] | 35.5 | 29.4 | 23.7 | 75.7 | 73.7 | 69.2 | 36.3 | 30.4 | 22.1 | 12.5 | 9.9 | 6.5 | 41.8 | 35.2 | 24.9 | 40.4 | 35.7 | 29.3 | 51.1 | 47.3 | 40.4 |
| | | DICS [23] | 34.0 | 23.1 | 15.7 | 72.1 | 70.1 | 63.6 | 34.8 | 22.3 | 13.1 | 12.2 | 7.3 | 4.5 | 35.5 | 18.5 | 9.3 | 37.7 | 28.3 | 21.2 | 48.0 | 38.8 | 32.2 |
| | Noisy LReID | DICS† [23] | 38.3 | 38.3 | 29.4 | 73.4 | 73.4 | 67.4 | 37.1 | 37.1 | 14.0 | 8.8 | 8.8 | 3.2 | 13.1 | 13.1 | 6.4 | 34.1 | 34.1 | 24.1 | 41.5 | 41.5 | 32.7 |
| | | CORE† [47] | 48.9 | **45.2** | 37.8 | **81.6** | 80.2 | 74.7 | 46.2 | 37.1 | 21.3 | **18.2** | 10.5 | 5.5 | 35.5 | 23.9 | 15.3 | 46.1 | 39.4 | 30.9 | 52.9 | 47.2 | 40.7 |
| | | LCNL†[45] | 28.9 | 23.5 | 16.2 | 69.2 | 67.9 | 63.1 | 29.4 | 20.0 | 11.6 | 9.7 | 6.8 | 4.3 | 38.1 | 27.8 | 18.6 | 35.1 | 29.2 | 22.8 | 45.1 | 38.4 | 33.8 |
| | | **CKP (Ours)** | 48.7 | 44.5 | **42.2** | 80.8 | **80.3** | **78.6** | **47.3** | **44.4** | **42.1** | 18.1 | **16.4** | **14.6** | **42.0** | **36.3** | **33.5** | **47.4** | **44.4** | **42.2** | **56.0** | **51.4** | **50.4** |
| R@1 | LReID | LwF [24] | **74.2** | 55.9 | 43.0 | 69.9 | 57.4 | 51.4 | 35.5 | 20.3 | 17.9 | 11.3 | 6.7 | 5.0 | 17.4 | 8.6 | 5.5 | 41.7 | 29.8 | 24.6 | 28.9 | 20.5 | 15.5 |
| | | PatchKD [34] | **74.2** | 56.9 | 49.5 | 70.5 | 61.7 | 56.0 | 34.1 | 23.1 | 18.1 | 11.0 | 7.2 | 5.5 | 17.2 | 9.8 | 5.9 | 41.4 | 31.7 | 27.0 | 29.6 | 21.7 | 18.9 |
| | | KRKC [50] | 54.0 | 42.0 | 34.4 | 75.1 | 68.8 | 63.4 | 38.4 | 28.0 | 19.8 | 14.9 | 9.8 | 7.5 | 37.1 | 19.6 | 11.1 | 43.9 | 33.6 | 27.2 | 36.9 | 29.4 | 24.3 |
| | | DKP [17] | 68.7 | 54.3 | 45.7 | 81.1 | 74.8 | 70.4 | 53.9 | 39.5 | 30.4 | 32.5 | 23.3 | 18.2 | 20.4 | 9.9 | 5.5 | 51.3 | 40.4 | 34.0 | 41.0 | 32.0 | 26.8 |
| | | LSTKC [42] | 66.8 | 60.5 | 50.2 | 81.3 | 77.6 | 70.9 | 59.3 | 34.0 | 19.5 | 35.5 | 16.2 | 8.8 | 27.7 | 14.6 | 6.2 | 54.1 | 40.6 | 31.1 | 40.3 | 30.7 | 25.0 |
| | LNL | CORE [47] | 63.0 | 55.8 | 49.8 | 78.7 | 77.0 | 72.6 | 56.7 | 48.9 | 38.8 | 32.9 | 26.9 | 19.0 | **43.1** | 36.2 | 24.1 | 54.9 | 49.0 | 40.9 | 44.7 | 41.7 | 34.5 |
| | | DICS [23] | 58.6 | 48.5 | 38.2 | 74.9 | 74.2 | 67.9 | 55.7 | 40.6 | 27.2 | 33.1 | 23.0 | 16.6 | 36.3 | 18.3 | 8.6 | 51.7 | 40.9 | 31.7 | 42.0 | 33.4 | 27.3 |
| | Noisy LReID | DICS† [23] | 65.6 | 65.6 | 56.4 | 76.8 | 76.8 | 70.9 | 57.3 | 57.3 | 28.1 | 26.7 | 26.7 | 12.4 | 12.8 | 12.8 | 6.2 | 47.8 | 47.8 | 34.8 | 34.9 | 34.9 | 27.3 |
| | | CORE† [47] | 72.6 | **69.9** | 62.8 | **83.9** | **83.0** | 77.4 | 63.8 | 55.5 | 37.5 | **41.0** | 27.1 | 16.8 | 36.4 | 22.9 | 14.7 | 59.5 | 51.7 | 41.8 | 46.0 | 40.9 | 35.3 |
| | | LCNL†[45] | 55.3 | 48.8 | 37.6 | 72.7 | 71.8 | 66.9 | 49.1 | 36.3 | 23.1 | 27.6 | 21.0 | 14.7 | 39.7 | 29.4 | 18.9 | 48.9 | 41.5 | 32.2 | 38.4 | 32.6 | 28.5 |
| | | **CKP (Ours)** | 71.8 | 68.1 | **66.9** | 83.2 | **83.0** | **81.0** | **64.7** | **62.1** | **58.9** | 40.1 | **37.5** | **34.7** | 42.4 | **37.1** | **34.1** | **60.4** | **57.6** | **55.1** | **49.5** | **44.6** | **43.4** |

ILR is not necessarily conducted after each epoch, and a rectification interval of $e_0$ epochs is adopted for computational efficiency.

**Model Training** During training, our framework can be integrated with existing LReID and LNL methods by introducing their noisy data learning loss $\mathcal{L}_{ReID}$ and our weighted knowledge distillation loss $\mathcal{L}_{wKD}$ (Equation (6) and (7)). Therefore, the overall loss is calculated by:

$$\mathcal{L} = \mathcal{L}_{ReID} + \mathcal{L}_{wKD}. \tag{9}$$

**Model Inference** We follow existing methods to use the feature generated by the final model $M_T$ for person matching.

## 4 EXPERIMENTS

### 4.1 Benchmark

In this paper, following the existing LReID [27] and Noisy ReID [47] works, a new Noisy LReID benchmark (NLReID) is proposed as below.

**Datasets**: NLReID contains 12 ReID datasets, 5 of them are used for lifelong training and evaluation (Market1501 [56], DukeMTMC-reID [30], CUHK-SYSU [41], MSMT17-V2 [39], and CUHK03 [21]), and the other 7 test datasets are used evaluate the generalizability of the model (CUHK01 [20], CUHK02 [19], VIPeR [8], PRID [11], i-LIDS [2], GRID [25], and SenseReID [53]). More details of the datasets are provided in our Supplementary Materials.

**Label Noise Generation**: Two different noisy settings are considered [47]. (1) *Random Noise* means a certain percentage (10%, 20%, 30%) of training data are randomly selected and assigned with random labels of other identities. (2) *Patterned Noise* means a certain percentage (10%, 20%, 30%) of randomly selected images are assigned with the labels of its most similar sample from other identities where the similarity is evaluated by a base model pre-trained with clean labels. Note that the random and patterned noises assume that the wrong labels are randomly distributed and semantic-relevant respectively. Usually, random noise can significantly disturb the learned feature but can be found out more easily. Whereas patterned noise shows a smaller influence on the learned feature but is harder to find out.

**Evaluation Metrics**: Following existing LReID works [6, 27, 34], the mean Average Precision (mAP) and Rank@1 accuracy (R@1) are adopted to evaluate the model performance on each seen and unseen dataset. Additionally, the seen/unseen average mAP and R@1 are reported to compare the lifelong learning and generalization capacity of the models across different scenarios.

### 4.2 Implementation Details

The state-of-the-art LReID method [42] is used as the baseline based on which our proposed CKR method is implemented. For

**Table 2: Results under the Patterned Noise. † indicates the state-of-the-art LNL method is combined with the latest anti-forgetting strategy of LSTKC.**

| Metric | Type | Method | Market-1501 | | | CUHK-SYSU | | | DukeMTMC | | | MSMT17 | | | CUHK03 | | | Seen-Avg | | | Unseen-Avg | | |
|---|---|---|---|---|---|---|---|---|---|---|---|---|---|---|---|---|---|---|---|---|---|---|---|
| | | | 10% | 20% | 30% | 10% | 20% | 30% | 10% | 20% | 30% | 10% | 20% | 30% | 10% | 20% | 30% | 10% | 20% | 30% | 10% | 20% | 30% |
| mAP | LReID | LwF [24] | 47.8 | 33.2 | 17.4 | 60.8 | 47.3 | 32.5 | 22.6 | 11.6 | 5.9 | 4.4 | 2.0 | 1.0 | 11.9 | 5.7 | 3.3 | 29.5 | 20.0 | 12.0 | 35.4 | 23.8 | 19.0 |
| | | PatchKD [34] | **50.6** | 34.2 | 18.9 | 62.1 | 48.2 | 34.7 | 22.1 | 12.1 | 5.9 | 4.5 | 2.2 | 1.1 | 11.4 | 5.5 | 2.9 | 30.1 | 20.4 | 12.7 | 35.1 | 25.5 | 18.5 |
| | | KRKC [50] | 31.9 | 22.3 | 18.0 | 73.1 | 66.7 | 63.5 | 27.1 | 16.2 | 10.4 | 5.7 | 3.4 | 2.6 | 39.1 | 24.8 | 16.1 | 35.4 | 26.7 | 22.1 | 45.4 | 36.9 | 31.9 |
| | | DKP [17] | 46.1 | 33.7 | 27.4 | 80.3 | 73.8 | 70.1 | 37.4 | 26.1 | 20.3 | 14.2 | 9.2 | 7.2 | 24.8 | 14.9 | 9.3 | 40.6 | 31.5 | 26.9 | 50.5 | 41.2 | 36.7 |
| | | LSTKC [42] | 40.3 | 36.0 | 33.2 | 78.3 | 75.4 | 71.6 | 38.3 | 30.6 | 18.0 | 15.2 | 8.0 | 4.4 | 30.2 | 20.0 | 13.2 | 40.5 | 34.0 | 28.1 | 50.8 | 42.2 | 37.9 |
| | LNL | CORE [47] | 35.7 | 33.4 | 30.1 | 75.8 | 74.8 | 73.3 | 36.9 | 30.9 | 29.2 | 13.5 | 11.1 | 9.3 | **42.4** | 37.7 | 31.5 | 40.9 | 37.6 | 34.7 | 52.2 | 46.5 | 45.0 |
| | | DICS [23] | 32.3 | 26.1 | 19.1 | 69.3 | 71.3 | 67.3 | 34.2 | 26.6 | 16.1 | 10.8 | 9.5 | 5.7 | 35.8 | 23.6 | 13.2 | 36.5 | 31.4 | 24.3 | 45.9 | 43.7 | 35.8 |
| | Noisy LReID | DICS† [23] | 40.5 | 39.5 | 31.4 | 77.4 | 75.0 | 68.1 | 40.1 | 39.5 | 33.8 | 16.4 | 9.0 | 6.9 | 33.7 | 21.1 | 16.4 | 41.6 | 36.8 | 31.3 | 52.7 | 47.4 | 41.1 |
| | | CORE† [47] | 49.0 | **49.6** | **45.4** | 80.8 | **80.0** | **79.0** | 46.4 | 42.2 | 38.7 | **19.2** | 9.2 | 10.8 | 39.5 | 32.2 | 21.7 | 47.0 | 42.6 | 39.1 | 55.8 | 48.6 | 47.6 |
| | | LCNL†[45] | 29.6 | 25.9 | 21.1 | 68.4 | 67.6 | 65.6 | 29.2 | 23.0 | 19.6 | 9.7 | 7.6 | 6.2 | 38.5 | 31.5 | 23.6 | 35.1 | 31.1 | 27.2 | 45.1 | 40.7 | 38.0 |
| | | **CKP (Ours)** | 50.1 | 46.9 | 44.1 | **81.0** | 79.9 | 78.9 | **47.2** | **45.2** | **43.1** | 18.3 | **17.1** | **15.7** | 41.9 | **39.9** | **36.6** | **47.7** | **45.8** | **43.7** | **57.3** | **54.4** | **51.1** |
| R@1 | LReID | LwF [24] | 72.0 | 59.8 | 37.5 | 63.8 | 49.3 | 32.7 | 42.1 | 25.0 | 14.5 | 14.5 | 7.0 | 4.4 | 12.8 | 6.7 | 3.8 | 41.0 | 29.6 | 18.6 | 28.3 | 17.8 | 13.7 |
| | | PatchKD [34] | **74.6** | 60.2 | 40.8 | 64.7 | 50.9 | 36.4 | 40.8 | 26.8 | 15.7 | 14.2 | 7.7 | 4.6 | 11.6 | 6.1 | 3.2 | 41.2 | 30.3 | 20.1 | 28.9 | 20.1 | 13.1 |
| | | KRKC [50] | 57.5 | 44.7 | 38.5 | 76.5 | 70.4 | 67.0 | 43.4 | 28.6 | 20.0 | 16.2 | 10.9 | 8.6 | 40.3 | 23.9 | 15.9 | 46.8 | 35.7 | 30.0 | 38.5 | 32.2 | 27.0 |
| | | DKP [17] | 71.1 | 60.2 | 53.8 | 82.9 | 77.2 | 74.1 | 55.4 | 42.8 | 36.8 | 34.8 | 25.9 | 21.5 | 24.8 | 14.3 | 7.6 | 53.8 | 44.1 | 38.8 | 43.4 | 35.9 | 30.7 |
| | | LSTKC [42] | 65.2 | 59.9 | 58.1 | 80.8 | 78.0 | 74.8 | 56.3 | 48.6 | 32.4 | 36.8 | 22.5 | 13.6 | 31.3 | 20.5 | 12.9 | 54.1 | 45.9 | 38.4 | 44.3 | 36.1 | 32.2 |
| | LNL | CORE [47] | 63.4 | 60.1 | 57.1 | 78.7 | 77.9 | 76.7 | 57.3 | 50.7 | 46.7 | 34.1 | 29.6 | 25.6 | **43.8** | 38.2 | 32.4 | 55.5 | 51.3 | 47.7 | 45.9 | 40.1 | 38.9 |
| | | DICS [23] | 57.3 | 51.9 | 43.2 | 72.5 | 75.0 | 71.3 | 55.1 | 44.5 | 31.2 | 29.7 | 28.4 | 19.3 | 36.4 | 24.1 | 13.0 | 50.2 | 44.8 | 35.6 | 38.7 | 37.3 | 30.3 |
| | Noisy LReID | DICS† [23] | 65.7 | 64.5 | 58.9 | 79.5 | 78.0 | 71.9 | 58.8 | 59.2 | 54.8 | 40.1 | 26.8 | 22.9 | 35.1 | 21.6 | 16.4 | 55.8 | 50.0 | 45.0 | 46.5 | 40.9 | 35.0 |
| | | CORE† [47] | 72.6 | **72.8** | **69.9** | **82.8** | **82.3** | **81.6** | 65.0 | 59.0 | 56.3 | **43.4** | 24.6 | 28.1 | 40.5 | 31.9 | 21.2 | 60.9 | 54.1 | 51.4 | 48.7 | 42.1 | 40.8 |
| | | LCNL†[45] | 55.8 | 51.6 | 47.0 | 71.5 | 71.1 | 69.5 | 49.2 | 41.1 | 35.7 | 27.2 | 23.0 | 19.4 | 39.6 | 33.3 | 24.6 | 48.7 | 44.0 | 39.2 | 38.4 | 34.9 | 32.3 |
| | | **CKP (Ours)** | 73.2 | 70.8 | 67.5 | 82.7 | **82.3** | 81.4 | **65.8** | **62.5** | **61.2** | 41.0 | **38.9** | **36.7** | 42.7 | **40.8** | **37.0** | **61.1** | **59.1** | **56.8** | **50.1** | **47.5** | **44.6** |

training, the first dataset is trained for 80 epochs and the subsequent datasets are trained for 60 epochs. 32 identities with 4 images for each identity are sampled as a mini-batch. The learning rate and weight decay are set as 0.008 and 0.0001 respectively, and an SGD optimizer is adopted. The hyperparameters $T_c$, $T_o$, $w_l$, $e_0$ are set to 0.8, 0.2, 0.1, 5 respectively.

## 4.3 The Comparison Methods

To comprehensively evaluate our method, extensive state-of-the-art LReID approaches (PatchKD [34], KRKC [50], LSTKC [42] and DKP [17]) are compared. The well-known class incremental learning method LwF [24] is also included. In addition, we combine the state-of-the-art LReID method, LSTKC, with the latest noisy label learning methods (LCNL [45], DICS [23], and CORE [47]) to adapt them to the Noisy LReID scenario. Note that parameter grid search is conducted to the above noisy LReID method to ensure their performance is optimized. All the methods above are implemented with the official codes, and we ensure a fair comparison by adopting the same backbone and data configurations.

## 4.4 Comparison with state-of-the-art methods

The results of different methods on the NLReID benchmark are reported in Table 1 and Table 2 under different ratios of random and patterned noise respectively. The best results under each scenario are highlighted in **Bold**.

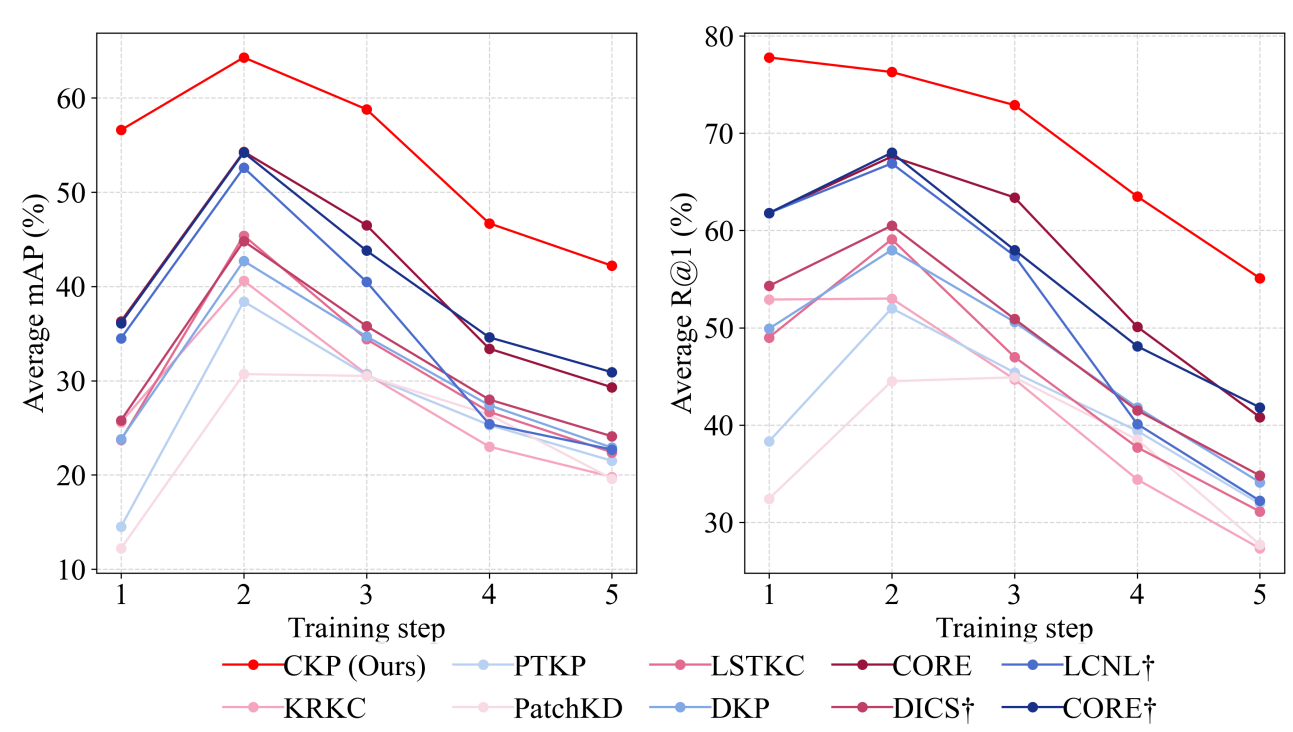

**Figure 3: The results of seen domain knowledge consolidation capacity under 30% random noise.**

**Compared to LReID Methods:** As is shown in Table 1 and Table 2, our model achieves significantly superior performance on the average performance of both seen and unseen domains compared to LReID methods since they are vulnerable to label

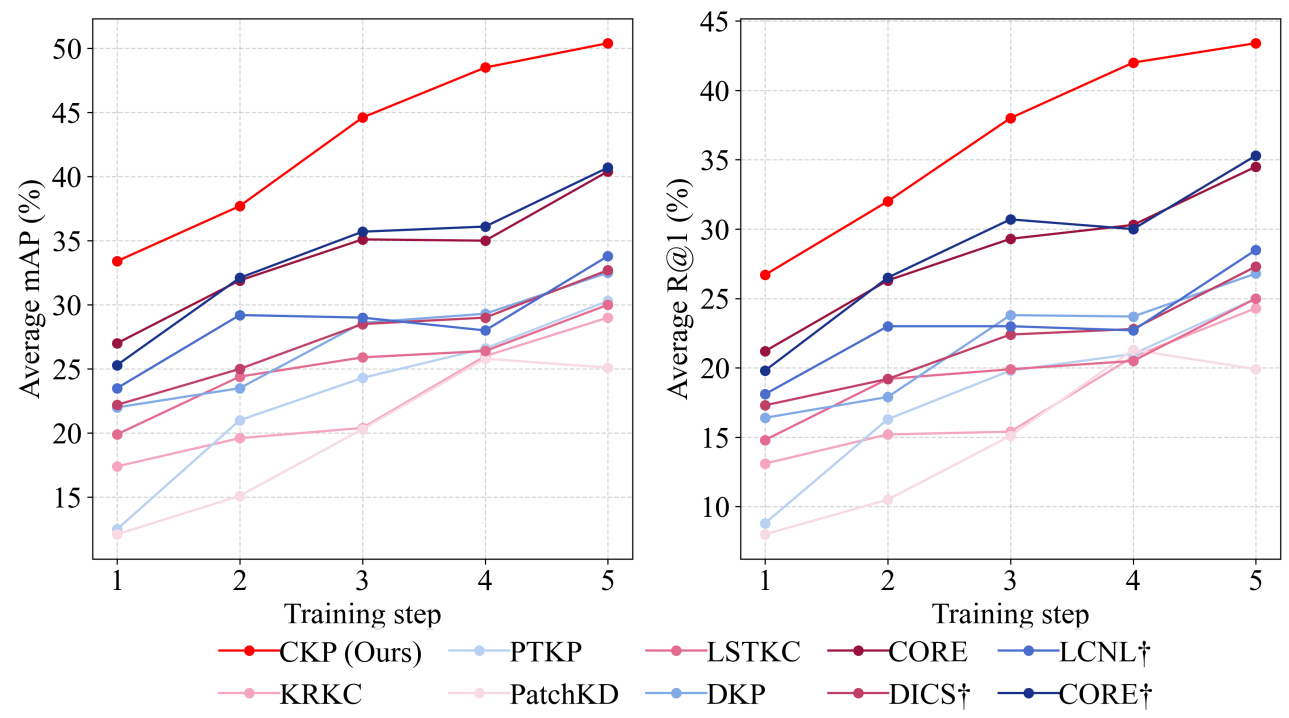

**Figure 4: The results of unseen domain generalization capacity under 30% random noise.**

noise. Specifically, when the random noise ratio increases from 10% to 30%, these methods exhibit degradation in average mAP and R@1 performance ranging from **12%-21%** under both seen and unseen domains. In contrast, our model experiences no more than a 6.1% degradation, attributed to the effectiveness of the new data purification and erroneous old knowledge filtering designs.

**Compared to LNL Methods:** As is shown in Table 1 and Table 2, the state-of-the-art LNL methods CORE and DICS achieve comparable results with LReID methods. But our CKP outperforms the better competitor CORE by a large margin. In particular, we achieve the average mAP/R@1 improvement of **12.9%/14.2%** and **10.0%/8.9%** under seen and unseen domains when learning under 30% random noise, and **12.4%/9.1%** and **10.0%/5.7%** improvement when learning under 30% patterned noise is obtained. The results arise because LNL methods are designed for stationary scenarios, neglecting the problems of catastrophic forgetting of correct knowledge and catastrophic remembering of erroneous knowledge.

**Compared to Noisy LReID Methods:** We incorporate the anti-forgetting strategy of the state-of-the-art LReID method LSTKC, into LNL methods, obtaining the Noisy LReID approaches DICS$^\dagger$, CORE$^\dagger$, and LCNL$^\dagger$. Among them, CORE$^\dagger$ exhibits the highest average performance across seen and unseen domains under both kinds of noises. As is shown in Table 1, compared to CORE$^\dagger$ under random noise, we achieve the improvement of **1.3%/0.9%**, **5.0%/5.9%**, and **11.3%/13.3%** on average mAP/R@1 performance in seen domains under noise ratios of **10%**, **20%**, and **30%**, respectively. Additionally, we also obtain the improvement of **3.1%/3.5%**, **4.2%/3.7%**, and **9.7%/8.1%** in the average mAP/R@1 performance of unseen domains under noise ratios of **10%**, **20%**, and **30%**, separately. As is shown in Table 2, under the challenging patterned noise, our CKP consistently obtains **0.7%/0.2%**, **3.2%/5.0%**, and **4.6%/5.4%** improvement on the average mAP/R@1 of seen domains under **10%**, **20%**, and **30%** noise. The increasing improvement under higher noise ratios highlights the superiority of our method in mining correct knowledge and reducing the remembering of erroneous knowledge in noisy scenarios.

**Seen Domain Performance Curves.** To show the new knowledge acquisition and anti-forgetting capacity of our model, We conduct experiments on the 30% random noise data in comparison with existing LReID and LNL methods. The results are shown in Figure 3. Compared to the competitors, our method outperforms them in the first dataset and maintains superiority throughout the training process. These results show that our proposed method could consistently consolidate correct knowledge by learning from the noise data of various domains.

**Unseen Domain Generalization Curves.** We further visualize the average performance on the unseen domains along the lifelong training steps, as depicted in Figure 4. The results demonstrate that our proposed model outperforms existing methods in capturing more generalizable knowledge when learning from non-stationary noisy data. This result is attributed to the knowledge purification mechanism of our model that ensures correct knowledge mining and erroneous knowledge filtering.

**Table 3: Ablation study of different components in CKR under 30% random noise.**

| Baseline | CDP | ILR | EKF | Seen-Avg mAP | Seen-Avg R@1 | Unseen-Avg mAP | Unseen-Avg R@1 |
|---|---|---|---|---|---|---|---|
| ✓ | | | | 30.9 | 41.8 | 40.7 | 35.3 |
| ✓ | ✓ | | | 38.8 | 51.6 | 47.3 | 40.2 |
| ✓ | | ✓ | | 34.5 | 45.7 | 43.5 | 37.4 |
| ✓ | | | ✓ | 34.8 | 46.1 | 43.8 | 37.3 |
| ✓ | ✓ | ✓ | | 41.9 | 54.7 | 49.2 | 43.1 |
| ✓ | ✓ | ✓ | ✓ | **42.2** | **55.1** | **50.4** | **43.4** |

## 4.5 Ablation Studies

In this section, we evaluate and discuss the effectiveness of our proposed components. All experiments are conducted on 30% random noise data in the NLReID benchmark.

**Ablations on different components**. In Table 3, we start with a CORE$^\dagger$ baseline and progressively integrate the proposed CDP, ILR, and EKF modules. The results illustrate that each module improves the model performance when utilized independently, and their combined utilization further boosts performance. Particularly noteworthy is the significant improvement yielded by CDP, underscoring the critical importance of ensuring training data clarity in mitigating label noise impact.

**Ablations on hyperparameters**. We analyze the effects of the hyperparameters $T_c$, $T_o$, $w_l$, and $e_0$, on the model in Figure 5. The results in Figure 5 (a) show that a relatively high $T_c$ helps improve the overall performance of the model, highlighting the importance of training data purity. In Figure 5 (b), we observe that an optimal $T_o$ tends to be relatively small, as some features may contain both correct and erroneous knowledge simultaneously. And $T_o$=0.2 shows the best balance. The results in Figure 5(c) suggest that a relatively low $w_l$ is optimal for label rectification, indicating that the model prediction is more reliable than the annotated label, yet the annotated label still contains some crucial clues that could remedy the imperfect predictions of the model. The results Figure 5 (d) show that frequently rectifying the label is not necessary and a rectification interval of 5 epochs is enough to guarantee the performance of the model. In practice, we set $T_c$, $T_o$, $w_l$, $e_0$ to 0.8, 0.2, 0.1, 5 respectively.

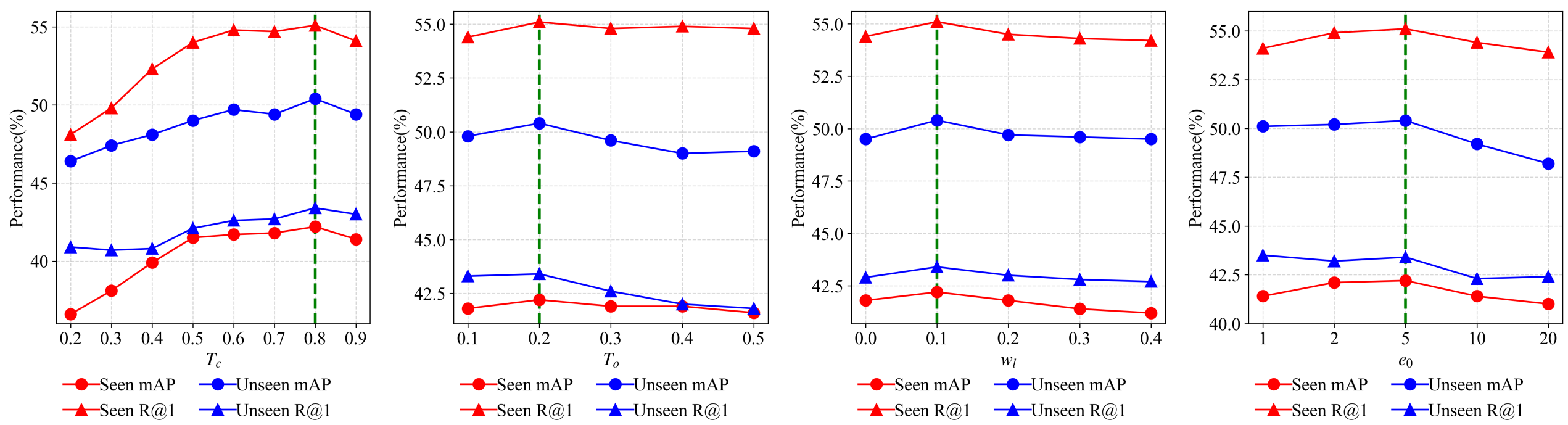

(a) label confidence threshold $T_c$ (b) feature confidence threshold $T_o$ (c) label rectification weight $w_l$ (d) label rectification interval $e_0$

**Figure 5: Ablation studies on hyperparameters under 30% random noise. Dashed green lines highlight our default values.**

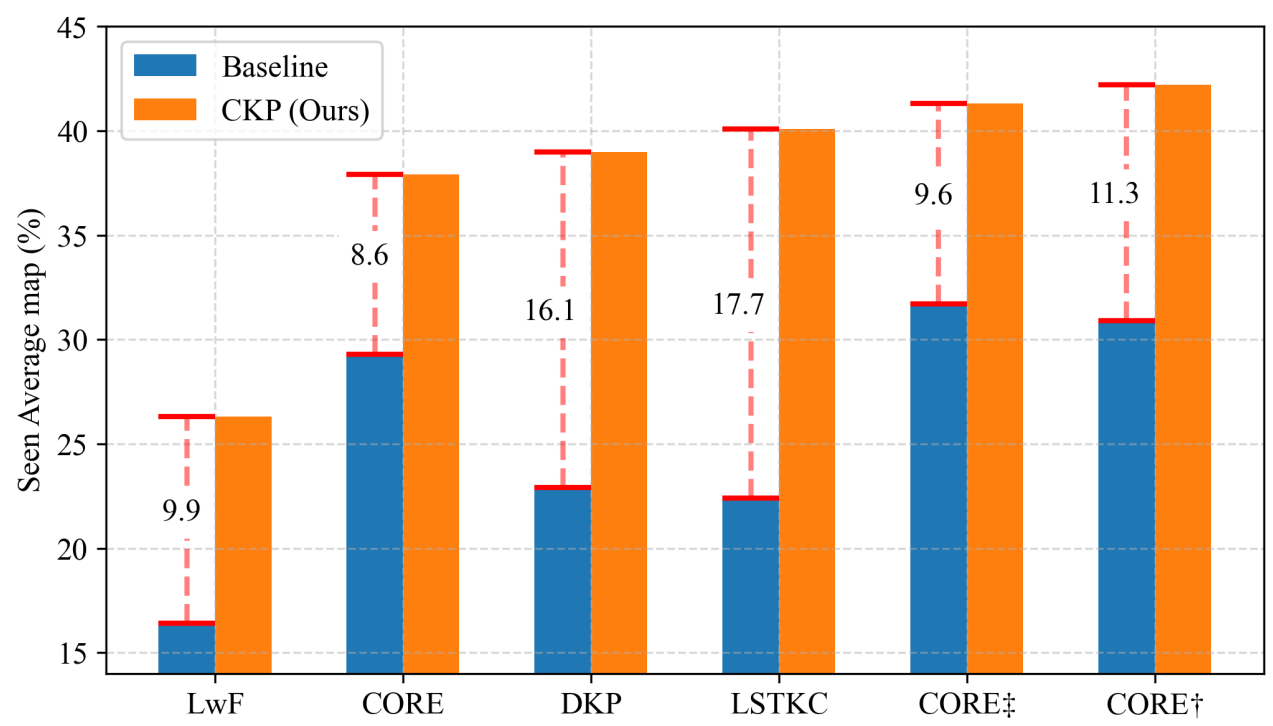

**Figure 6: The proposed CKR can be readily integrated with existing LReID and LNL methods to significantly improve their Noisy LReID performance.**

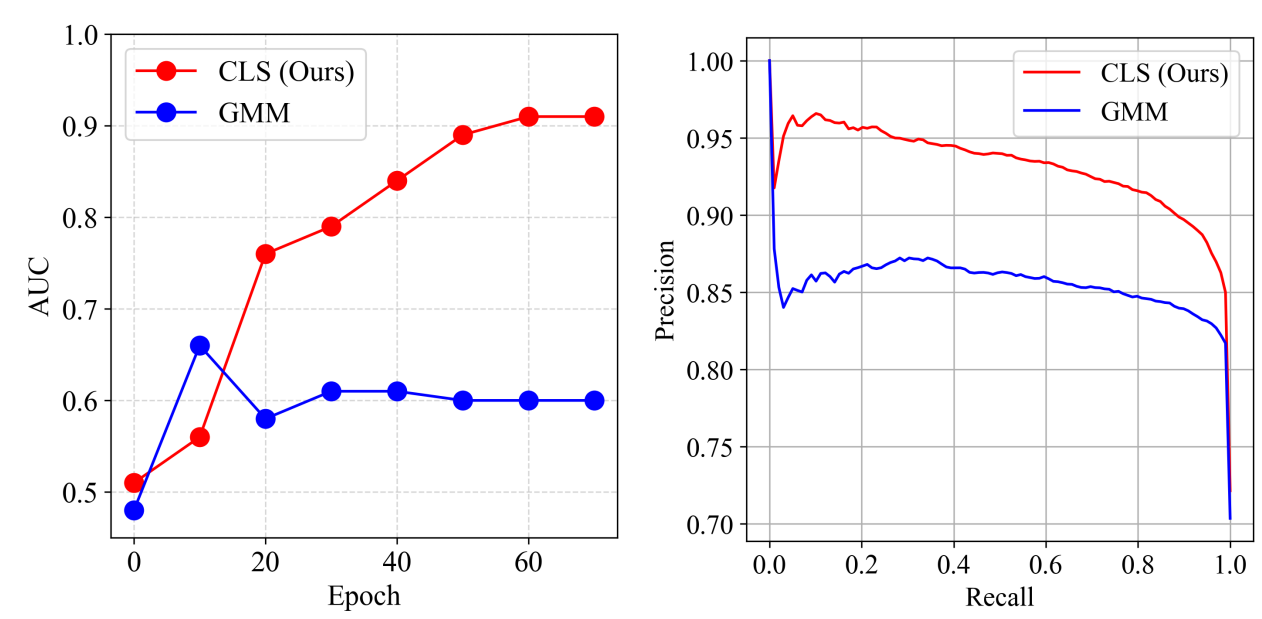

(a) Label scoring AUC tendency. (b) Model prediction ROC curve.

**Figure 7: Effectiveness of proposed CLS compared to GMM under 30% random noise.**

**Combination with other methods**. As is shown in Figure 6, when our method is combined with existing methods, considerably within 8.6%-17.7% improvement is achieved. CORE$^\dagger$ and CORE$^\ddagger$ represent integrating CORE with the anti-forgetting strategy of LSTKC and DKP, respectively. Note that LSTKC and DKP are inter-instance relation-based knowledge distillation methods and LwF is logits-based knowledge distillation method. These results demonstrate the compatibility of our method with different anti-forgetting strategies and our method can improve the Noisy LReID performance of existing methods consistently.

**Effectiveness of Cluster-aware Label Scoring**. To evaluate the confidence estimation capability of our CLS strategy which plays an important role in our CDP and EKF, we experimentally replace CLS with the widely-used Gaussian Mixture Model (GMM) in our approach. Figure 7 (a) illustrates the tendency of label scoring AUC across training epochs. Initially, GMM performs slightly better, but as the model begins to overfit the label noise, its label scoring capacity diminishes after the 10th epoch. In contrast, our CLS consistently improves its AUC performance and surpasses GMM after the 20th epoch. This shows the superiority of our CLS in guiding the algorithms to collect clean data. Furthermore, we visualize the ROC curves of the predicted identities generated by the final models in Figure 7 (b). It is evident that our CLS effectively guides the model to learn the correct knowledge.

## 5 CONCLUSION

In this paper, we initially investigate a practical task Noisy Lifelong Person Re-Identification (Noisy LReID), which suffers exacerbated correct knowledge catastrophic forgetting and additional erroneous knowledge catastrophic remembering problems. To facilitate research in Noisy LReID, we introduce a benchmark named NLReID. In addition, we propose a novel and effective Continual Knowledge Purification (CKP) framework. To reduce the erroneous knowledge acquisition, an Iterative Label Rectification pipeline and a Cluster-aware Data Purification module are designed to rectify the noise labels and collect clean data along the training procedure to mitigate the influence of noisy data on new knowledge learning. Besides, to handle the catastrophic remembering and forgetting issues, an Erroneous Knowledge Filtering algorithm is proposed to reduce erroneous old knowledge accumulation and ensure correct knowledge consolidation. Extensive experiments show our method is robust to different kinds of label noise and achieves significant Noisy LReID performance improvement, especially under high-ratio noise compared to existing methods.

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
