# OpenReview forum: "Mitigate Catastrophic Remembering via Continual Knowledge Purification for Noisy Lifelong Person Re-Identification"
_acmmm.org/ACMMM/2024/Conference — MM2024 Poster_

### Official Review · Reviewer_UsKh · 2024-05-20

**Rating:** 3
**Confidence:** 3

**Summary:**

This paper proposes a new noisy lifelong person re-identification scheme, and subsequently designs a Continual Knowledge Purification (CKP) method to address the problem. Firstly, a Cluster-aware Data Purification module is designed to select high-confidence data for training. Besides, the wrong labels are rectified through fusing the prediction and label information. Finally, a Erroneous Knowledge Filtering (EKF) algorithm is proposed to only transfer the correct old knowledge. The proposed method achieves state-of-the-art performance at the noisy lifelong person re-id benchmark.

**Strengths:**

- The writing is clear and easy to follow.
- The proposed method demonstrates substantial improvement at the noisy lifelong re-id benchmark.
- The method design is reasonable where only effective knowledge is accumulated throughout the lifelong learning pipeline.

**Limitations:**

1. This work proposes a new noisy lifelong re-id scheme. However, there exist several issues on the scheme setting and the proposed method.
- Given that recent unsupervised re-id methods already achieve performance similar to supervised ones, why not directly use unsupervised methods for lifelong learning? There are plenty of unsupervised methods leveraging confidence or certainty to filter out outlier samples from training, some examples of which are listed below. The proposed CLS and CDP modules adopt a similar strategy to improve the quality of training data. On the one hand, the contribution is marginal. On the other hand, unsupervised methods naturally avoid the influence of noisy labels. When the noise ratio is high, it would be interesting to see the performance achieved by unsupervised methods.
    - [1] Exploiting Sample Uncertainty for Domain Adaptive Person Re-Identification.
    - [2] Self-paced contrastive learning with hybrid memory for domain adaptive object re-id.
    - [3] Multi-view evolutionary training for unsupervised domain adaptive person re-identification.
    - [4] Deep credible metric learning for unsupervised domain adaptation person re-identification.
- In Tab. 1, it can be observed that the influence of noise is significantly larger than lifelong learning. By only applying noisy learning methods, the performance is already higher than naive lifelong learning methods even under a low noise ratio (10%). Tab. 3 also shows that the improvement is largely based on the CDP, which is a module designed for dealing with noise. It then brings up the following question: how does the proposed method perform on standard noisy learning benchmarks?
2. More specific examples (like images) are expected to see the label purification and rectification effects.
3. Some expression details require further refinement:
(1) line 362; (2) line 395.

**Suitability:**

2

---

### Official Review · Reviewer_1Wi6 · 2024-05-24

**Rating:** 3
**Confidence:** 3

**Summary:**

The paper addresses the problem of lifelong person re-identification (LReID) in the presence of noisy data with incorrect labels, which poses a challenge due to the model's tendency to remember erroneous knowledge and the issue of catastrophic forgetting. The authors propose a Continual Knowledge Purification (CKP) method to tackle these challenges and introduce a Noisy LReID benchmark for performance evaluation. Experimental results demonstrate that the CKP method achieves state-of-the-art performance in handling noisy data and addressing the catastrophic remembering and forgetting issues in LReID.

**Strengths:**

This paper addresses an important problem of lifelong noise person re-identification.
The paper is well-organized and clearly written.

**Limitations:**

The contribution of this paper's method is not very clear. As discussed in Section 1, lines 139-140, directly applying LNR methods to lifelong noisy person re-identification faces the challenge of "neglecting the disparity between the prediction/loss and the actual identity distribution." However, from the method design presented in this paper, it is unclear how the authors address this challenge.

**Suitability:**

2

---

### Official Review · Reviewer_AWTf · 2024-05-25

**Rating:** 2
**Confidence:** 4

**Summary:**

This paper proposes a new task, i.e., Noisy Lifelong Person Re-Identification. And to solve this task, the authors propose a Continual Knowledge Purification (CKP) method to address the catastrophic remembering of erroneous knowledge and catastrophic forgetting of correct knowledge simultaneously. The CKP method first uses a clustering method to select clean data to re-train the model, and then try to transfer the correct old knowledge to the new model with a EKF module.

**Strengths:**

Through experiments, the proposed method is verified effective in the Noisy Lifelong Person Re-Identification setting.

**Limitations:**

1.	The necessity of proposing a new Noisy Lifelong Person Re-Identification task is not adequately justified. Is there a real-world demand for this? Since LReID only requires minimal training data, is the probability of noise presence in these few data sets relatively low?
2.	The expression in the third paragraph of the Introduction is not clear. For instance, in line 136, "these approaches neglect the disparity between the prediction/loss and the actual identity distribution" - what is this disparity, and how is it neglected? In line 146, "directly applying these LNL methods to Noisy LReID without considering the characteristics of LReID" - what characteristics of LReID are not considered?
3.	Limited novelty of the proposed model.
4.	The model name is noy unified, “CKP” in Abstract and “CKR” in Introduction.
5.	Is there a necessity to conduct experiments using the proposed method in the general LReID setting?
6.	In Figure 1, the image labeled x4 on the right side of panel (a) with ID 2 seems to be a correct example. The x-axis labels in panels (b) and (c) are not explained.
7.	Comparing the last two rows of the results in Table 3, the REF seems to promote very little performance improvement, only 0.3 R@1 in Unsee-Avg.

**Suitability:**

1

---

### Official Review · Reviewer_78JQ · 2024-05-26

**Rating:** 4
**Confidence:** 2

**Summary:**

This work proposes a new task of lifelong person re-identification with noisy labels (Noisy LReID). By analyzing the main challenges in Noisy LReID, the authors propose an effective Continual Knowledge Rectification method for filtering noisy data and continuously learning correct knowledge.

**Strengths:**

This work for the first time proposes to explore the Noisy LReID. The proposed method is shown to be superior in tackling this problem.

**Limitations:**

1. I'm curious about the impact of the noisy labels in LReID. For example, comparing the performances of previous LReID methods when the noise ratio is equal to 0 and greater than 0; visualizing the feature distributions of the same LReID model learned from noisy and clean datasets, etc.

2. Any experiments to verify the accuracy of CDP and ILR? For example, how many noisy samples are accurately filtered out by CDP? How many noisy labels are accurately rectified by ILR? How does the accuracy of ILR change during the continual learning procedure?

3. Is it applicable to integrate some of the proposed modules in previous methods to further verify their effectiveness?

**Suitability:**

2

---

### Meta-Review · Area_Chair_PPd9 · 2024-07-01

**Recommendation:** Accept (Poster)
**Confidence:** 5

**Metareview:**

The paper initially received the following ratings: BR (1Wi6), WR (AWTf), BR (UsKh), and BA (78JQ). After rebuttal and discussion, the final ratings were: BR (1Wi6), BA (AWTf), WA (UsKh), and BA (78JQ).

Below are the details of the rebuttal:
- Reviewer AWTf and Reviewer UsKh were mostly satisfied with the response and ultimately raised their ratings.
- Reviewer 78JQ was satisfied with the rebuttal and maintained a borderline acceptance rating.
- Reviewer 1Wi6 retained the original borderline rejection rating without additional justification.

The AC carefully reviewed the rebuttal:
1) For the contribution of technique, the AC agrees that although the techniques in the proposed method are not novel, they effectively address a new and challenging task.
2) For the contribution of task setting, while Reviewers AWTf, UsKh, and 78JQ initially questioned the necessity of the proposed Noise LReID setting, they were satisfied with the response after reading the rebuttal and subsequently raised their ratings.
3) Regarding improvement, as a more challenging task, the AC considers the improvements acceptable.

In sum, after the rebuttal, the AC believes the concerns raised by the reviewers were well-addressed and recommends acceptance of this paper. In the final version, the AC strongly encourages the authors to include all discussions from the rebuttal and improve the presentation to enhance readability for a diverse audience.

---

### Meta-Review · Senior_Area_Chairs · 2024-07-10

**Recommendation:** Accept (Poster)
**Confidence:** 4

**Metareview:**

This paper received mixed ratings initially. After rebuttal, most reviewers are satisfied with the reponse. SAC and AC carefully checked the reviews and rebuttal and recommend acceptance of the paper.